# Delayed inhibition mechanism for secondary channel factor regulation of ribosomal RNA transcription

Sarah K Stumper[1], Harini Ravi[1], Larry J Friedman[1], Rachel Anne Mooney[2], Ivan R Corrêa Jnr[3], Anne Gershenson[4], Robert Landick[2,5], Jeff Gelles[1]*

[1]Department of Biochemistry, Brandeis University, Waltham, United States; [2]Department of Biochemistry, University of Wisconsin, Madison, United States; [3]New England Biolabs, Ipswich, United States; [4]Department of Biochemistry and Molecular Biology, University of Massachusetts, Amherst, United States; [5]Department of Bacteriology, University of Wisconsin, Madison, United States

**Abstract** RNA polymerases (RNAPs) contain a conserved 'secondary channel' which binds regulatory factors that modulate transcription initiation. In *Escherichia coli*, the secondary channel factors (SCFs) GreB and DksA both repress ribosomal RNA (rRNA) transcription, but SCF loading and repression mechanisms are unclear. We observed in vitro fluorescently labeled GreB molecules binding to single RNAPs and initiation of individual transcripts from an rRNA promoter. GreB arrived and departed from promoters only in complex with RNAP. GreB did not alter initial RNAP-promoter binding but instead blocked a step after conformational rearrangement of the initial RNAP-promoter complex. Strikingly, GreB-RNAP complexes never initiated at an rRNA promoter; only RNAP molecules arriving at the promoter without bound GreB produced transcript. The data reveal that a model SCF functions by a 'delayed inhibition' mechanism and suggest that rRNA promoters are inhibited by GreB/DksA because their short-lived RNAP complexes do not allow sufficient time for SCFs to dissociate.

*For correspondence:
gelles@brandeis.edu

## Introduction

Multi-subunit RNA polymerases (RNAPs), the molecular machines responsible for the synthesis of cellular mRNAs, contain catalytic subunits that are conserved across all domains of life (*Cramer, 2002*; *Werner and Grohmann, 2011*). In addition to their conserved catalytic machinery, RNAPs also contain conserved binding sites for regulatory proteins. An important such regulatory hot spot is the secondary channel, a pore that provides access to the deeply-buried active site of the enzyme (*Nickels and Hochschild, 2004*). Multiple regulators called secondary channel factors (SCFs) compete to control the activity of RNAPs by binding in the secondary channel. Two evolutionarily distinct SCF families have evolved in bacteria and eukaryotes, distinguished by insertion into the channel of either antiparallel α-helices (bacteria; GreA-like proteins) or of a β hairpin (eukaryotes, TFIIS-like proteins). *Escherichia coli* uses at least five distinct GreA-like SCFs (*Zenkin and Yuzenkova, 2015*). Although the *E. coli* SCFs have related three-dimensional structures (*Stepanova et al., 2009*), they have distinct effects on transcription; in addition, some *E. coli* SCFs can act at multiple points in the transcription cycle. GreB, for example, was first identified as an SCF that stimulates endonucleolytic cleavage of the nascent transcript in transcription elongation complexes (ECs). GreB stimulates cleavage of RNA that has reverse threaded through the active site (backtracked) by multiple nucleotides, for instance after nucleotide misincorporation at the RNA 3′ end (*Borukhov et al., 2001*; *Borukhov et al., 1993*; *Laptenko et al., 2003*). By causing removal of the backtracked segment of the nascent transcript both GreB and its paralog, GreA, can stimulate transcription by aiding

promoter escape (*Duchi et al., 2016*; *Feng et al., 1994*; *Hsu et al., 1995*; *Lerner et al., 2016*; *Shimamoto et al., 1986*; *Susa et al., 2002*) and by allowing escape from promoter-proximal σ70-paused ECs (*Deighan et al., 2011*; *Marr and Roberts, 2000*).

In contrast, GreB (but not GreA) also can *inhibit* initiation by RNAP prior to phosphodiester bond formation at certain promoters with short-lived RNAP-promoter complexes such as the ribosomal RNA (rRNA) *rrnB* P1 promoter (*Rutherford et al., 2007*). This GreB activity is similar to one of the activities of a different *E. coli* SCF, DksA, which has a divergent domain that binds outside the secondary channel. DksA does not promote transcript cleavage and has other effects on transcript elongation (*Zenkin and Yuzenkova, 2015*; *Zhang et al., 2014*), but it can strongly affect initiation either negatively at 'stringent' promoters (e.g., *rrnB* P1) or positively at other promoters (*e.g.*, the histidine biosynthetic operon promoter P$_{hisG}$) (*Gourse et al., 2018*; *Paul et al., 2005*; *Paul et al., 2004*; *Potrykus et al., 2006*; *Rutherford et al., 2007*). Production of rRNA is suppressed during the stringent response, a global reprogramming of gene expression in response to amino acid starvation (*Browning and Busby, 2016*). DksA binds in the SC and reduces initiation from *rrnB* P1 during the stringent response (*Paul et al., 2004*; *Ross et al., 2016*; *Rutherford et al., 2009*; *Rutherford et al., 2007*); this effect is potentiated by binding of guanosine tetraphosphate (ppGpp), which is accumulated at concentrations up to ~1 mM to signal amino acid starvation. ppGpp binds two sites on RNAP to modulate transcription: one at the interface of the omega and β′ subunits and a second created by DksA upon binding in the SC (*Molodtsov et al., 2018*; *Ross et al., 2016*). There are multiple proposals for mechanisms by which DksA and ppGpp suppress *rrnB* P1 initiation in the stringent response (*Paul et al., 2005*; *Rutherford et al., 2009*; *Rutherford et al., 2007*).

Although much study has focused on stringent regulation of rRNA production, cells must also maintain appropriate levels of rRNA during rapid-growth (i.e., non-starvation) conditions, conditions in which rRNAs are the most abundant transcripts produced. A growth rate-dependent control (*Bartlett and Gourse, 1994*) system maintains rRNA levels relative to total cellular protein by making rRNA synthesis proportional to the steady-state growth rate rather than to amino acid availability (*Gourse et al., 1986*; *Josaitis et al., 1995*). This regulation occurs at micromolar cellular concentrations of ppGpp (*Ross et al., 2016*; *Ryals et al., 1982*) rather than the millimolar concentrations that mediate stringent regulation. In contrast to stringent regulation of rRNA transcription (in which DksA is clearly implicated), there is potential for involvement of both GreB and DksA in growth rate dependent control (*Mallik et al., 2006*; *Rutherford et al., 2007*). The mechanism for growth rate-dependent control of rRNA synthesis by the two SCFs is unknown, and the point in the transcription cycle at which the SCFs load onto RNAP has not been defined.

To elucidate the mechanism(s) of rRNA initiation regulation by DksA and GreB, it is essential to define how these SCFs load onto RNAP and modulate the kinetics of *rrnB* P1 promoter complexes at different stages in the process of initiation. Here we used both fluorescence correlation spectroscopy (FCS) and colocalization single-molecule spectroscopy (CoSMoS) (*Friedman et al., 2006*), a multiwavelength single-molecule fluorescence microscopy technique, to directly observe and quantify the dynamic interactions in vitro of fluorescently labeled GreB molecules with individual RNAP molecules and *rrnB* P1 promoter•RNAP complexes. We also visualized the inhibition of individual RNAP molecules by GreB and DksA in the presence and absence of the micromolar concentrations of ppGpp corresponding to those present in rapidly growing (non-stringent) cells. Our results suggest that GreB binds to core RNAP and σ70RNAP holoenzyme an order of magnitude more tightly than previously thought and that GreB/DksA competition for binding to RNAP free in the cytoplasm likely determines the relative importance of these two factors in modulating rRNA initiation under rapid growth conditions. Using GreB as a model to investigate the mechanism(s) of SCF•RNAP binding and action at *rrnB P1*, we reveal the 'delayed inhibition' mechanism for SCF loading onto RNAP and suggest how short-lived promoter-RNAP complexes are selectively inhibited by SCFs.

## Results

### GreB tightly binds RNAP and σ70RNAP

Although the dynamics of GreB interactions with ECs have been directly observed and quantitatively characterized (*Furman et al., 2013*; *Tetone et al., 2017*), analogous measurements of GreB dynamics with core RNAP and σ70RNAP holoenzyme are lacking. To begin investigating the effects of GreB

on transcription initiation, we first measured the equilibrium dissociation constant ($K_D$) for the GreB•$\sigma^{70}$RNAP complex (*Figure 1A*) using fluorescence correlation spectroscopy (FCS). The experiments used a fluorescently labeled GreB$^{Cy3}$ construct shown to be fully active in transcript cleavage assays (*Tetone et al., 2017*). The large size difference between GreB and RNAP clearly distinguishes the FCS autocorrelation curves of free GreB$^{Cy3}$ (fast-diffusing species) from $\sigma^{70}$RNAP•GreB$^{Cy3}$ complex (slow-diffusing species) (*Figure 1B*). The autocorrelation curves for free GreB$^{Cy3}$ were fit to obtain a diffusion constant of $60 \pm 10$ $\mu m^2$ $s^{-1}$ (all uncertainties are SE unless otherwise indicated). Autocorrelation curves obtained for $\sigma^{70}$RNAP yielded (see Materials and methods) the diffusion constant of GreB$^{Cy3}$ bound to $\sigma^{70}$RNAP (10–20 $\mu m^2$ $s^{-1}$) and fraction GreB$^{Cy3}$ bound. A hyperbolic fit to the fraction bound data at varying $\sigma^{70}$RNAP concentrations revealed tight binding with $K_D = 10 \pm 1$ nM (*Figure 1C*). Similar experiments performed with core RNAP again yielded $K_D = 10 \pm 1$ nM (*Figure 1D*). These measurements demonstrate that GreB binds both core and $\sigma^{70}$RNAP an order of magnitude more tightly than was estimated in previous non-equilibrium measurements of binding (*Furman et al., 2013*; *Loizos and Darst, 1999*; *Rutherford et al., 2007*) and an order of magnitude more tightly than previously measured for GreB binding to ECs (*Tetone et al., 2017*; *Table 1*). The dissociation constant for the interaction is also ~10 fold tighter than the values reported for DksA in the absence of ppGpp ($K_D = 101 \pm 19$ or 112 [no S.E. reported] nM for core, $K_D = 99 \pm 15$ or 52 [no S.E. reported] nM for $\sigma^{70}$ holoenzyme) (*Lennon et al., 2009*; *Molodtsov et al., 2018*). GreB$^{Cy3}$ binding to $\sigma^{70}$RNAP was quantitatively indistinguishable from that of unlabeled GreB (see Materials and methods), indicating that the high binding affinity is not an artefact of dye labeling.

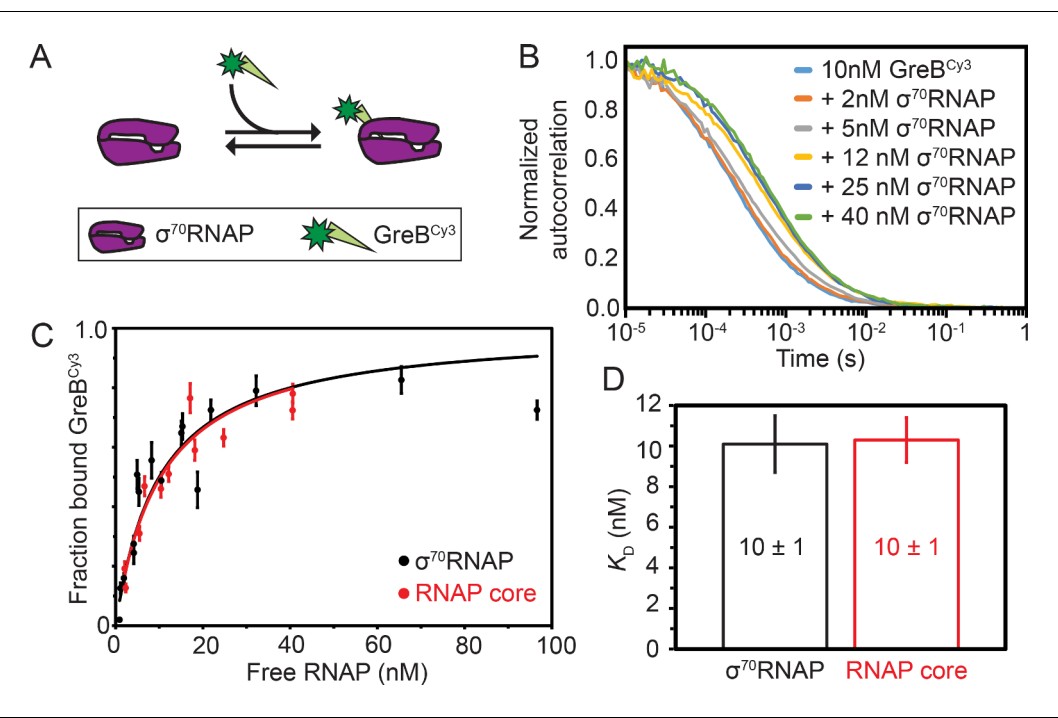

**Figure 1.** Thermodynamic and kinetic stability of GreB•RNAP complexes. (**A**) Experimental scheme for measuring by FCS the affinity with which Cy3- (green star) labeled GreB binds $\sigma^{70}$RNAP. (**B**) Examples of normalized FCS autocorrelation curves obtained after adding the indicated amount of $\sigma^{70}$RNAP to GreB$^{Cy3}$. (**C**) Fraction of GreB$^{Cy3}$ bound to RNAP determined at different $\sigma^{70}$RNAP (black) and RNAP core (red) concentrations in the presence of 10 nM GreB$^{Cy3}$, determined by fitting FCS autocorrelation curves including those shown in (**B**). Fraction bound measurements (points; ±SE) were fit to a Langmuir binding equation (lines). (**D**) Dissociation equilibrium constants $K_D$ (±SE) determined from the fits in (**C**).

The online version of this article includes the following figure supplement(s) for figure 1:

**Figure supplement 1.** Kinetics of GreB dissociation from RNAP and $\sigma^{70}$RNAP.

**Figure supplement 2.** Kinetics of GreB binding to and dissociation from RNAP and $\sigma^{70}$RNAP.

To measure the kinetics of GreB-polymerase interactions, we collected FCS data at different time points after initiating the dissociation of GreB$^{Cy3}$•σ$^{70}$RNAP complexes by addition of 100-fold excess unlabeled wild-type GreB (*Figure 1—figure supplement 1*). Dissociation of GreB$^{Cy3}$ from σ$^{70}$RNAP was observed, but the rate of dissociation was comparable to the time scale of FCS data collection, making reproducible measurement difficult. Therefore, we performed CoSMoS single-molecule fluorescence equilibrium kinetics experiments (*Friedman and Gelles, 2015*) with tethered σ$^{70}$RNAP or core RNAP to measure GreB$^{Cy3}$ binding and dissociation rates (*Figure 1—figure supplement 2*). These experiments facilitated comparison with measurements on ECs using a similar method (*Table 1*; *Tetone et al., 2017*). The single-molecule data showed evidence of a minor subpopulation (~10%) of polymerase molecules with distinct kinetics (see Discussion). Nevertheless, all data were consistent with the conclusions that core RNAP and σ$^{70}$RNAP interact similarly with GreB$^{Cy3}$, that binding is very fast (of order $10^7$ M$^{-1}$ s$^{-1}$, close to the diffusion limit for molecules of these sizes), and that dissociation was 0.2 s$^{-1}$ or slower. The $K_D$ values derived from the kinetic parameters for the surface-tethered RNAP are close to those measured in solution by FCS (*Table 1*), indicating that tethering does not greatly alter binding affinity. In additional experiments, we saw that wild-type GreB and the single-cysteine mutant rcGreB effectively compete with GreB$^{Cy3B}$ for binding to tethered σ$^{70}$RNAP. Binding kinetics for all three proteins were indistinguishable, indicating that the GreB mutation and labeling do not affect binding to σ$^{70}$RNAP (*Figure 1—figure supplement 2C*). We also observed that GreB$^{Cy3B}$ lifetimes were not limited by photobleaching under the conditions of the experiments (*Figure 1—figure supplement 2G*). Taken together, these kinetic and equilibrium data show that even though GreB is principally viewed as an elongation factor, it binds to σ$^{70}$RNAP roughly an order of magnitude more tightly than to ECs and its dissociation is at least an order of magnitude slower (*Table 1*; *Furman et al., 2013*; *Tetone et al., 2017*). Thus, in the cell σ$^{70}$RNAP holoenzyme is expected to effectively compete with ECs for GreB binding if pools of GreB are limiting.

## The tight-binding interaction of GreB or DksA with σ$^{70}$RNAP inhibits transcription initiation from *rrnB* P1

Near-saturating (2 µM) concentrations GreB and DksA equally inhibited initiation in vitro from the rRNA promoter *rrnB* P1 and inhibition by both proteins was equally enhanced by 100 µM ppGpp (*Rutherford et al., 2007*). However, these experiments followed only initiation reaction endpoints, not rates, complicating mechanistic interpretation. Furthermore, an SCF may be capable of multiple

**Table 1.** Kinetics of GreB$^{Cy3B}$ interactions with RNAPs.

| Experiment | Species | Time to first binding measurement | | | Dwell time distribution fit parameters[†] | $K_D$ (nM) |
| | | $k_{bkgnd}$ ($10^7$ M$^{-1}$ s$^{-1}$) | $k_{on}$ ($10^7$ M$^{-1}$ s$^{-1}$) | $A_f$[‡] | | |
|---|---|---|---|---|---|---|
| FCS (*Figure 1*) | RNAP core | N.D. | N.D. | N.D. | N.D. | 10 ± 1 |
| FCS (*Figure 1*) | σ$^{70}$RNAP | N.D. | N.D. | N.D. | N.D. | 10 ± 1 |
| CoSMoS (*Figure 1—figure supplement 2*) | RNAP core | 0.00023 ± 0.00009 (N = 12) | 0.94 ± 0.06 (N = 163) | 0.87 ± 0.13 | $a = 91 ± 4\%$ $\tau_1 = 5.3 ± 0.5$ $\tau_2 = 27 ± 3$ s (N = 872) | 18 ± 2* |
| CoSMoS (*Figure 1—figure supplement 2*) | σ$^{70}$RNAP | 0.0017 ± 0.0008 (N = 29) | 1.5 ± 0.9 (N = 152) | 0.43 ± 0.11 | $a = 89 ± 7\%$ $\tau_1 = 5.6 ± 0.6$ $\tau_2 = 25 ± 7$ s (N = 943) | 11 ± 1* |
| CoSMoS[§] | EC | N.R. | 2.7 ± 1.0 | 0.80 ± 0.06 | $\tau = 0.4 ± 0.1$ s | 93 ± 36 |
| CoSMoS[§] | Static EC−6 | N.R. | 1.9 ± 0.1 | 1 | $\tau = 0.29 ± 0.02$ s | 184 ± 8 |

*Calculated as $K_D = (k_{on} \tau_1)^{-1}$.

[†]From single- or bi-exponential fit.

[‡]Active fraction of surface-tethered molecules (*Friedman and Gelles, 2015*).

[§]Data from *Tetone et al. (2017)*.

N.D., not determined; N.R., not reported.

modes of interaction with RNAP (*Molodtsov et al., 2018*; *Zenkin, 2014*) and these different interactions may have a range of affinities. To ask whether the high affinity (DksA, $K_D$ =~100 nM) or very high affinity (GreB, $K_D$ =~10 nM) binding of the proteins to σ$^{70}$RNAP inhibits *rrnB* P1 initiation, we measured the rate of the overall initiation pathway (i.e., starting with free holoenzyme binding to DNA) at low SCF concentrations equal to these $K_D$s. In a CoSMoS experiment (*Friedman and Gelles, 2012*), template DNA molecules with *rrnB* P1 were tethered to the surface of an observation chamber, and transcription was initiated by introducing σ$^{70}$RNAP and NTPs (*Figure 2A*). The reaction mixture also contained two TAMRA-labeled oligonucleotide probes complementary to transcript nucleotides 1–27 and 29–54. Initiation was detected as co-localization of a spot of green-excited TAMRA fluorescence with a spot of blue-excited fluorescence from the AF488 dye coupled to the DNA template (*Figure 2B* and *Figure 2—figure supplement 1*). As expected, probe co-localization was dependent on the presence of NTPs and RNAP. Measured times to first probe binding on each DNA molecule were fit with an exponential model (*Figure 2C–F*), yielding the apparent first-order initiation rate constants (*Table 2*; see Materials and methods).

Addition to the experiment of 10 nM GreB reduced the rate of initiation approximately four-fold (*Figure 2C*, purple vs. green curves; *Figure 2G*, leftmost two bars). This substantial inhibition in rate contrasts with the only 8% reduction of the initiation end point seen at even higher (100 nM) GreB concentrations (*Rutherford et al., 2007*). Significantly, the inhibitory effect of DksA is identical to that of GreB (within experimental uncertainty) when we performed the analogous experiment with DksA at its $K_D$ in place of GreB. (*Figure 2D*; *Figure 2G*). Thus, GreB and DksA have identical effects on the rate of the full initiation process when the effects are measured at the same fractional occupancy of σ$^{70}$RNAP. We found similarly indistinguishable inhibitory effects of GreB and DksA in the presence of 100 μM ppGpp (*Figure 2E–G*; *Rutherford et al., 2007*). Thus, GreB inhibits initiation ~10 fold more potently than DksA on a concentration basis, but the difference can be fully accounted for by the difference in the binding affinities of the two proteins, without reference to any functional differences between GreB- and DksA-bound RNAP.

## GreB arrives at and departs from an rRNA promoter in complex with σ$^{70}$RNAP

Since SCF occupancy of holoenzyme paralleled inhibition of rrnB initiation, we next asked at which stage(s) of the *rrnB* initiation process the SCFs bind to RNAP. We simultaneously examined binding of 1 nM σ$^{70}$RNAP$^{647}$ (a fluorescently-labeled holoenzyme preparation; see Materials and methods) and 10 nM GreB$^{Cy3B}$ to the surface-tethered *rrnB* P1 DNA. The experiments (*Figure 3A*) were conducted without NTPs, which allows for promoter complex formation but not initiation (*Rutherford et al., 2009*). As expected, we observed transient co-localization of σ$^{70}$RNAP$^{647}$ molecules with individual template molecules, reflecting formation of promoter complexes (*Figure 3B*, red). Control experiments demonstrated that the observed lifetimes of these complexes were not appreciably limited by photobleaching (*Figure 3—figure supplement 1*). Separate control experiments in the presence of NTPs confirmed that transcription initiation kinetics with GreB$^{Cy3B}$ and/or σ$^{70}$RNAP$^{647}$ are similar to those seen with the unlabeled proteins (*Table 2*).

In these three-color CoSMoS experiments, we observed that some σ$^{70}$RNAP$^{647}$ molecules arrived at DNA locations in complex with GreB$^{Cy3B}$ molecules (orange highlights in *Figure 3B* and *Figure 3—figure supplement 2*), whereas others arrived without detectable GreB$^{Cy3B}$ fluorescence (purple highlights). Even though essentially all GreB$^{Cy3B}$ molecules are labeled (see Materials and methods), this result is nevertheless expected since the experiment is conducted at 10 nM GreB$^{Cy3B}$, the concentration at which only 50% of the polymerase molecules in solution have GreB bound. The fraction of σ$^{70}$RNAP$^{647}$ molecules observed to arrive simultaneously with GreB$^{Cy3B}$ agrees with that expected (*Figure 3C*), demonstrating that σ$^{70}$RNAP•GreB$^{Cy3B}$ and σ$^{70}$RNAP have similar rate constants for binding to DNA.

Of the holoenzyme molecules that arrived at the promoter without bound GreB, very few recruited GreB after arrival. In particular, at 10 nM GreB$^{Cy3B}$ we saw only 3.5 ± 0.9% of σ$^{70}$RNAP•DNA complexes bind GreB (47 instances of binding to 1,355 complexes over a summed duration of 1,121 s, corresponding to $k_{on}$ = (0.42 ± 0.06)×10$^7$ M$^{-1}$ s$^{-1}$). This binding is specific – it is much larger than the rate constant for background GreB$^{Cy3B}$ binding to DNA in the absence of σ$^{70}$RNAP (32 binding events recorded on 168 DNA molecules each observed for 1,342 s, corresponding to $k_{on}$ = (1.4 ± 0.3)×10$^4$ M$^{-1}$ s$^{-1}$) – and is comparable to the rate constant for GreB

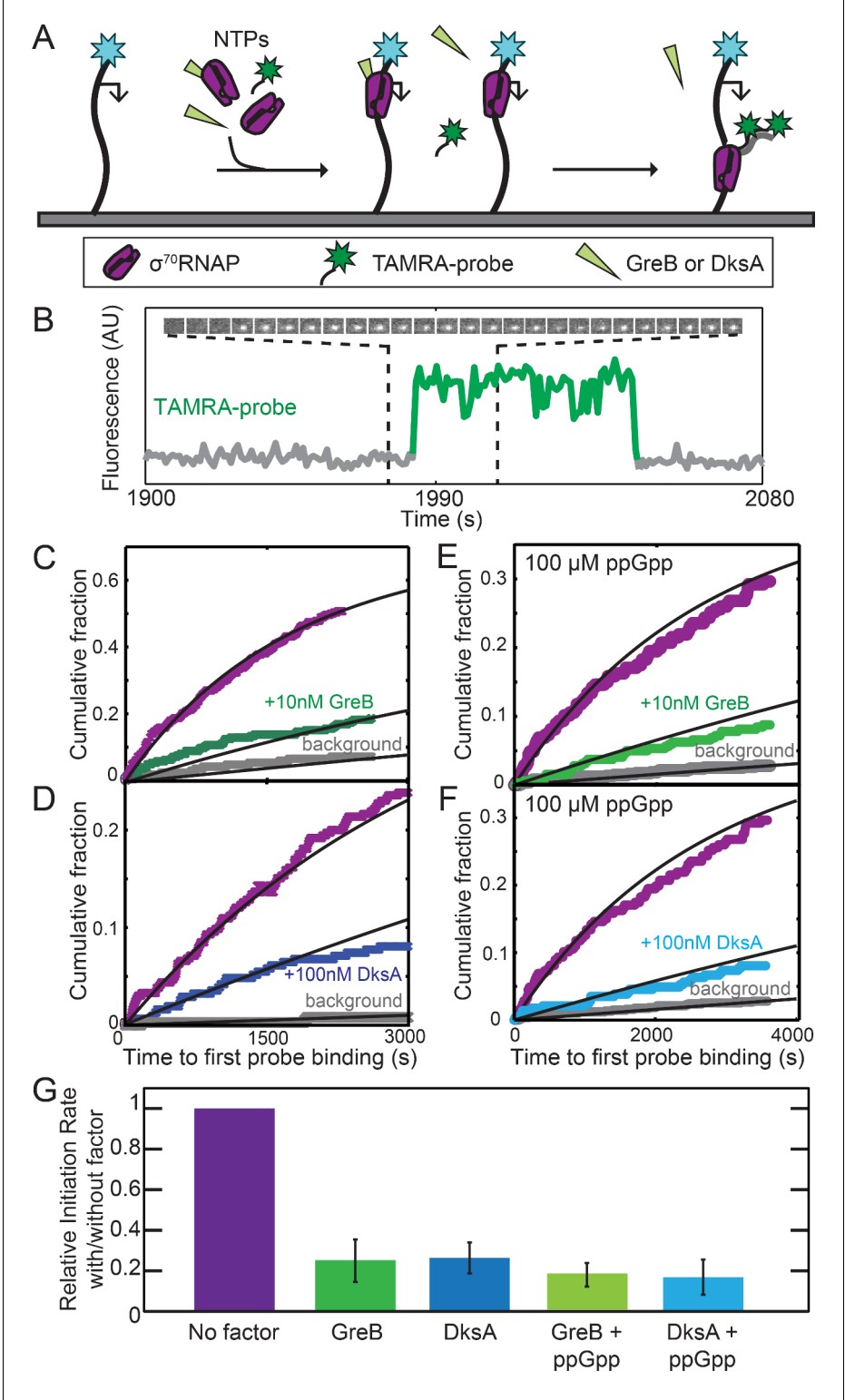

**Figure 2.** GreB and DksA equally inhibit transcription initiation from *rrnB* P1 at equal occupancies of σ⁷⁰RNAP. (**A**) Schematic of CoSMoS transcription initiation rate measurement in the presence of GreB or DksA. DNA molecules (314 bp) containing the *rrnB* P1 promoter (bent arrow) and labeled with AF488 (blue star) were tethered to the chamber surface via a biotin-neutravidin linkage. Initiation in the presence or absence of DksA, GreB, and/or ppGpp is detected as co-localization of two oligonucleotide probes labeled with TAMRA (green star) that hybridize near the 5′ end of the nascent transcript. (**B**) Example transcript probe fluorescence images (1.1 × 1.1

*Figure 2 continued on next page*

*Figure 2 continued*

µm) taken at 5 s intervals (top) and corresponding intensity record (bottom) from the location of a single template DNA molecule. An objective image analysis algorithm (see Materials and methods) detected a spot of probe fluorescence at the times shown in green. (C–F) Cumulative distributions of the time until first probe detection on each DNA molecule. Only probe colocalization events 5 s or longer in duration were scored. Measurements were made in the absence (purple) or presence (green and blue) of secondary channel factors GreB (C, E) or DksA (D, F), without (C, D) or with (E, F) 100 µM ppGpp. Exponential fits (lines) are corrected (see Materials and methods) for non-specific binding of probe to the chamber surface measured at control locations that lack DNA molecules (gray). (G) Inhibition of initiation by secondary channel factors with and without 10 nM GreB, 100 nM DksA, and/or 100 µM ppGpp derived from experiments in (C–F), taken from the fit parameters (*Table 2*). Graph indicates the rate of initiation in the presence of the indicated factors (blue and green) relative to that measured in the same experiment without the factors (purple). Error bars: S.E.

The online version of this article includes the following figure supplement(s) for figure 2:

**Figure supplement 1.** Additional examples of time records of transcript probe (green-excited) fluorescence co-localized at single *rrnB* P1 DNA locations.

---

binding to holoenzyme (*Table 1*). However, given the short lifetime of rRNA promoter complexes this rate is still too small to allow more than a minor fraction of $\sigma^{70}$RNAP•DNA complexes to recruit GreB even at concentrations of GreB several times the 10 nM used here. At the higher concentrations essentially all holoenzyme would already have bound GreB (since $K_D$ = 10 nM), which would also eliminate the recruitment pathway. Thus these data suggest that GreB suppression of *rrnB* P1 initiation is essentially completely controlled by GreB binding to $\sigma^{70}$RNAP off DNA rather than GreB binding to initiation complexes, and that this conclusion holds in both high (>>$K_D$) and low (<<$K_D$) GreB concentration regimes.

Just as GreB arrived at the promoter in complex with holoenzyme, we saw that GreB departed from DNA when, and only when, $\sigma^{70}$RNAP dissociated (e.g., *Figure 3B* and *Figure 3—figure supplement 2*). More than 90% of recorded $\sigma^{70}$RNAP•GreB•DNA complexes ended with simultaneous departure of GreB$^{Cy3B}$ and $\sigma^{70}$RNAP$^{647}$ (*Figure 3D*, orange highlight). The few non-simultaneous events seen occurred with a frequency not much higher than expected from coincidental colocalization or photobleaching, as estimated by a randomized control analysis of the same data (*Figure 3D*, line). Thus, we see no evidence for GreB dissociation that leaves behind polymerase bound to the promoter. The observations that GreB nearly always arrives at the promoter together with RNAP and only dissociates with RNAP suggest that the extent of GreB inhibition of *rrnB* P1 initiation is quantitatively set by the amount of GreB holoenzyme occupancy.

## DNA binding rates and closed complex lifetimes of $\sigma^{70}$RNAP are unaffected by GreB or DksA

We next investigated how GreB or DksA bound to $\sigma^{70}$RNAP affect the initial processes in initiation: formation, interconversion, and dissociation of promoter complexes. In our experiments, which use unmodified *rrnB* P1 in a linear DNA in the absence of NTPs, the promoter complexes formed are almost entirely in either the initial bound state or an early closed-complex intermediate. Later, open complex states form transiently under these conditions, but they are likely too short-lived to be resolved in the time resolution of our experiments (*Barker et al., 2001*; *Haugen et al., 2008*; *Rutherford et al., 2009*).

Even at concentrations at which half of the holoenzyme molecules are occupied, the CoSMoS data show that neither SCF significantly altered $k_a$, the second-order rate constant for initial binding of $\sigma^{70}$RNAP$^{647}$ to *rrnB* P1 DNA (*Figure 3E*; *Table 3*). This conclusion was confirmed by direct comparison of the DNA binding rates of the subsets of $\sigma^{70}$RNAP$^{647}$ molecules that either were or were not seen to have bound GreB$^{Cy3B}$ (*Figure 3—figure supplement 3*; *Table 3*). Since GreB and DksA do not slow $\sigma^{70}$RNAP binding, inhibition of initiation must be caused by effects on other step(s) of the initiation pathway. However, we also saw no substantial alteration of $\sigma^{70}$RNAP$^{647}$ dwell times on *rrnB* P1 when GreB or DksA was present (*Figure 3F*; *Table 3*). Therefore, kinetic destabilization of the two resolvable closed complexes also cannot account for the observed inhibition of initiation from *rrnB* P1. Taken together, these data imply that the SCFs act at a step in initiation after

**Table 2.** Fit parameters for single molecule initiation experiments*.

| Conditions | Active fraction | $N^†$ | $k_{bkgnd}$ ($10^{-5}$ s$^{-1}$) | $k_{init}$ ($10^{-4}$ s$^{-1}$) | Avg $k_{init}$ ($10^{-4}$ s$^{-1}$) (Fold inhibition)‡ |
|---|---|---|---|---|---|
| $\sigma^{70}$RNAP | 0.63 ± 0.12 | 42/225 | 1.1 ± 0.4 | 6.1 ± 0.6 | 5.8 ± 0.6 (1) |
| | 0.37 ± 0.08 | 55/253 | 1.3 ± 0.2 | 6.4 ± 0.3 | |
| | 0.29 ± 0.05 | 90/373 | 0.9 ± 0.4 | 5.6 ± 0.5 | |
| | 0.47 ± 0.09 | 143/307 | 0.9 ± 0.6 | 4.9 ± 0.8 | |
| | 0.68 ± 0.10 | 185/298 | 0.6 ± 0.3 | 3.1 ± 0.5 | |
| | 0.62 ± 0.14 | 124/201 | 1.1 ± 0.9 | 7.0 ± 0.9 | |
| | 0.87 ± 0.13 | 279/322 | 7.7 ± 0.2 | 8.5 ± 0.3 | |
| $\sigma^{70}$RNAP +10 nM rcGreB | 0.63 ± 0.12 | 19/278 | 1.1 ± 0.4 | 1.5 ± 0.3 | 1.3 ± 0.4 (4.4 ± 0.7) |
| | 0.47 ± 0.09 | 24/252 | 0.9 ± 0.6 | 1.2 ± 0.6 | |
| $\sigma^{70}$RNAP +10 nM GreB (*Figure 2C*) | 0.62 ± 0.14 | 21/246 | 2.6 ± 0.6 | 1.2 ± 0.2 | 1.9 ± 0.5 (3.1 ± 0.9) |
| | 0.87 ± 0.13 | 16/198 | 7.7 ± 0.2 | 2.3 ± 0.4 | |
| $\sigma^{70}$RNAP +10 nM GreB +100 µM ppGpp (*Figure 2E*) | 0.37 ± 0.08 | 18/276 | 1.3 ± 0.2 | 1.1 ± 0.4 | 1.0 ± 0.7 (5.8 ± 1.1) |
| | 0.47 ± 0.09 | 14/231 | 0.9 ± 0.6 | 0.9 ± 0.3 | |
| $\sigma^{70}$RNAP +100 nM DksA (*Figure 2D*) | 0.29 ± 0.05 | 25/302 | 0.9 ± 0.4 | 0.6 ± 0.3 | 1.7 ± 0.6 (3.4 ± 1.0) |
| | 0.68 ± 0.10 | 36/305 | 0.6 ± 0.3 | 1.7 ± 0.8 | |
| $\sigma^{70}$RNAP +100 nM DksA + 100 µM ppGpp (*Figure 2F*) | 0.37 ± 0.08 | 14/202 | 1.3 ± 0.2 | 1.3 ± 0.5 | 1.1 ± 0.6 (5.3 ± 1.0) |
| | 0.68 ± 0.10 | 24/299 | 0.6 ± 0.3 | 1.0 ± 0.6 | |
| $\sigma^{70}$RNAP$^{647}$ | 0.71 ± 0.19 | 265/377 | 0.7 ± 0.4 | 5.3 ± 0.8 | 5.5 ± 0.7 (0.95 ± 0.6) |
| | 0.49 ± 0.13 | 104/215 | 0.8 ± 0.3 | 5.8 ± 0.5 | |
| $\sigma^{70}$RNAP$^{647}$ +10 nM GreB$^{Cy3B}$ | 0.71 ± 0.19 | 29/306 | 0.7 ± 0.4 | 1.2 ± 0.9 | 1.4 ± 0.7 (3.9 ± 1.0) |
| | 0.49 ± 0.13 | 14/147 | 0.8 ± 0.3 | 1.6 ± 0.6 | |
| $\sigma^{70}$RNAP$^{647}$ +10 nM GreB$^{Cy3B}$ +100 µM ppGpp | 0.71 ± 0.19 | 12/228 | 0.7 ± 0.4 | 0.9 ± 0.5 | 0.9 ± 0.5 (6.1 ± 0.7) |
| | 0.49 ± 0.13 | 17/311 | 0.8 ± 0.3 | 0.9 ± 0.4 | |

*At least two replicates per experimental condition were performed due to variability in active fraction between trials. The rate of TAMRA-probe binding to AF488-DNA locations ($k_{init}$) and rate of TAMRA-probe binding to locations without AF488-DNA molecules ($k_{bkgnd}$) were obtained for each individual replicate. The mean initiation rate (Avg $k_{init}$) was then calculated for each set of experimental conditions.

†The two numbers given for each replicate are the number of *rrnB* P1 DNA molecules on which a TAMRA-probe binding lasting >5 s was observed and the total number of DNA molecules in that replicate.

‡ Fold-inhibition was determined by dividing the Avg $k_{init}$ for the relevant RNAP ($\sigma^{70}$RNAP or $\sigma^{70}$RNAP$^{647}$) by the calculated Avg $k_{init}$ in the presence of the specified SCF/ppGpp. For the $\sigma^{70}$ RNAP$^{647}$ alone condition, fold inhibition is calculated relative to unlabeled $\sigma^{70}$RNAP.

formation of these initial closed complexes, such as isomerization to open complex or subsequent nucleotide addition reactions.

## $\sigma^{70}$RNAP does not initiate rRNA transcription while GreB is bound

To further investigate the mechanism of GreB inhibition of *rrnB* P1 initiation, we next performed experiments in which we monitored $\sigma^{70}$RNAP$^{647}$ binding, GreB$^{Cy3B}$ binding, and initiation on the same DNA molecule. Since this required separate observation of four different fluorescently labeled molecules, we used an experimental design in which we first recorded the locations of AF488-labeled DNA molecules and then photobleached them so that we could later record the hybridization to nascent RNA of an oligo probe labeled with the same dye (*Figure 4A*). In these initiation experiments conducted in the presence of NTPs and GreB, we observed that a small fraction of RNAP binding events lead to transcript production (39 of 1,755 total RNAP binding events from two replicate experiments; examples are shown in *Figure 4B* and *Figure 4—figure supplement 1*). Strikingly, in none (zero of 39) of these productive binding events did RNAP arrive accompanied by

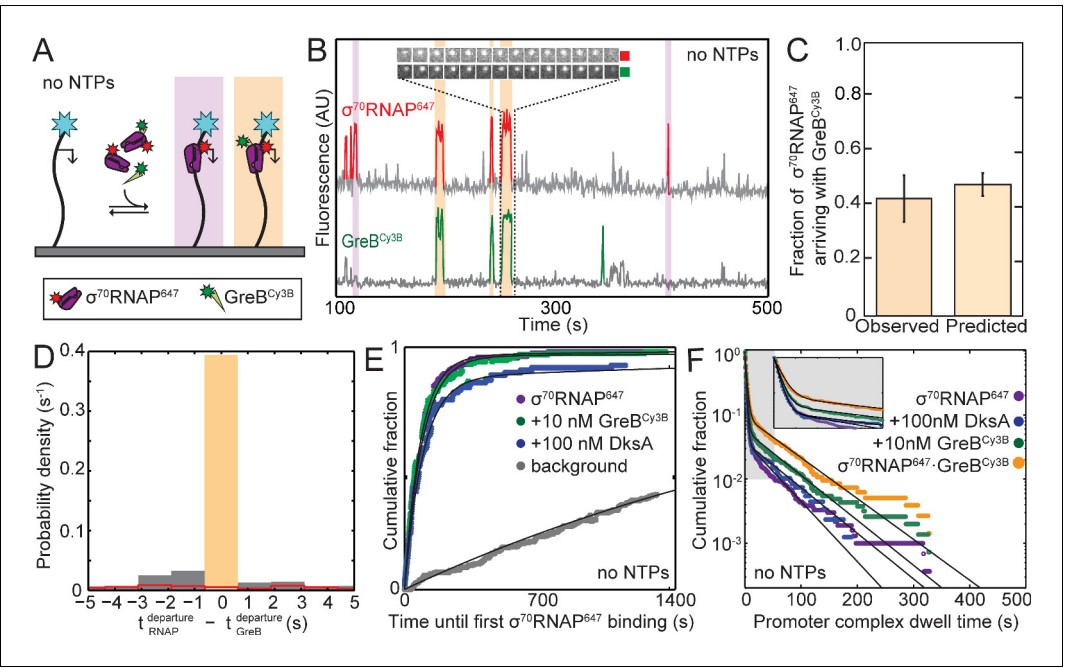

**Figure 3.** GreB and RNAP arrive together at *rrnB* P1 and depart together. (**A**) Experiment design. Experiments were conducted in the absence of NTPs to monitor colocalization of σ[70]RNAP[647] (1 nM) and GreB[Cy3B] (10 nM) with the same surface-tethered *rrnB* P1 DNA used in *Figure 2*. (**B**) Recording excerpt of fluorescence co-localized with a single DNA molecule. Colored portions of the traces indicate detection of a DNA-colocalized spot of σ[70]RNAP[647] (red) or GreB[Cy3B] (green) fluorescence. Events in which molecules of σ[70]RNAP[647] and GreB[Cy3B] simultaneously arrive at and later simultaneously depart from the DNA molecule (orange) and events in which σ[70]RNAP[647] arrives alone (purple) are highlighted. Inset shows time series images (0.5 s per frame; each 1.1 × 1.1 μm) of one of these events. (**C**) Fraction (±SE) of σ[70]RNAP[647] DNA binding events (*N* = 2,408) in which a GreB[Cy3B] arrived simultaneously (±1 frame) with σ[70]RNAP[647] (left) and the fraction predicted by an equilibrium model (see Materials and methods) in which σ[70]RNAP and GreB·σ[70]RNAP bind to the promoter with equal association rate constants (right). (**D**) Histogram of time differences between σ[70]RNAP[647] departure and GreB[Cy3B] departure from σ[70]RNAP[647]•GreB[Cy3B]•DNA complexes (solid bars). Only complexes that formed by simultaneous arrival (C; *N* = 1,053) were analyzed. Departure was simultaneous (±1 frame; orange highlight) in 91% of complexes. In a control analysis of the same data (red, open bars) in which departure events from different molecules were randomly paired (see Materials and methods), less than 3% were simultaneous. Plot shows the center of the distributions; 6% of experimental data and 61% of control fall outside the plotted range. In (**C**) and (**D**), data are aggregated from three replicate experiments. (**E**) Cumulative fraction of DNA molecules to which the first binding of σ[70]RNAP[647] occurred at or before the indicated time. Experiments were conducted with or without SCFs as indicated. Background curve (gray) is collected from randomly selected locations without detected DNA molecules. (For clarity, only the background curve from the experiment without SCFs is shown). Exponential fits (lines), each corrected using background data taken from the same recording yielded RNAP-DNA association rate constants (*Table 3*). (**F**) Cumulative distributions of σ[70]RNAP[647] dwell times on *rrnB* P1 DNA in the absence or presence of added SCFs (points). Also shown is the distribution of the subset of σ[70]RNAP[647] dwell times observed in the presence of 10 nM GreB[Cy3B] for which GreB[Cy3B] arrived and departed simultaneously with σ[70]RNAP[647] (orange). Inset: magnified view. Fits to a biexponential dwell time distribution (black) yielded parameters reported in *Table 3*.

The online version of this article includes the following figure supplement(s) for figure 3:

**Figure supplement 1.** Photobleaching control for σ[70]RNAP[647].

**Figure supplement 2.** Fluorescence intensity records from six additional DNA locations demonstrating σ[70]RNAP[647] and GreB[Cy3B] co-localization in the absence of NTPs.

**Figure supplement 3.** Effect of bound GreB[Cy3B] on the kinetics of σ[70]RNAP[647] association with DNA.

GreB. In contrast, RNAP arrived together with GreB in hundreds (~50% of 1,755 total RNAP binding events) of non-productive RNAP binding events. Just as was observed in the absence of NTPs (*Figure 3C,D*) RNAP molecules that arrived on the DNA with GreB retained bound GreB for the

**Table 3.** Fit parameters for time to first RNAP binding (**Figure 3E** and **Figure 3—figure supplement 3**) and RNAP dwell time (**Figure 3F**) distributions on *rrnB* P1 DNA[*].

| Experimental conditions | Time to first binding measurement | | | Dwell time measurement |
|---|---|---|---|---|
| | $k_{bkgnd}$ ($10^6$ M$^{-1}$s$^{-1}$) | $k_a$ ($10^6$ M$^{-1}$s$^{-1}$) | $A_f$ | |
| $\sigma^{70}$RNAP$^{647}$ (**Figure 3E**) | 0.037 ± 0.006 (N = 34) | 2.2 ± 0.6 (N = 227) | 0.95 ± 0.04 | a = 91 ± 4% $\tau_1$ = 2.6 ± 0.4 s $\tau_2$ = 29.7 ± 0.2 s (N = 2,576) |
| +10 nM GreB$^{Cy3B}$ (**Figure 3E**) | 0.016 ± 0.002 (N = 19) | 2.0 ± 0.4 (N = 269) | 0.93 ± 0.06 | a = 91 ± 6% $\tau_1$ = 3.1 ± 0.7 s $\tau_2$ = 42.2 ± 0.4 s (N = 2,408) |
| +100 nM DksA (**Figure 3E**) | 0.0093 ± 0.0005 (N = 8) | 1.9 ± 0.5 (N = 187) | 0.89 ± 0.05 | a = 92 ± 7% $\tau_1$ = 2.5 ± 0.4 s $\tau_2$ = 29.9 ± 0.8 s (N = 1,381) |
| +10 nM GreB$^{Cy3B}$; $\sigma^{70}$RNAP$^{647}$· GreB$^{Cy3B}$ subset[†] (**Figure 3—figure supplement 3**) | 0.016 ± 0.002 (N = 19) | 2.4 ± 0.7 (N = 124) | 0.94 ± 0.05 | a = 90 ± 5% $\tau_1$ = 3.4 ± 0.8 s $\tau_2$ = 44.6 ± 0.7 s (N = 1,053) |
| +10 nM GreB$^{Cy3B}$; $\sigma^{70}$RNAP$^{647}$ subset[†] (**Figure 3—figure supplement 3**) | 0.016 ± 0.002 (N = 19) | 1.8 ± 0.6 (N = 145) | 0.91 ± 0.08 | N.D. |

[*]**Figure 3E and F** each show curves from an individual experimental replicate; fit parameters here represent pooled data from all three replicates.

[†]Data in 10 nM GreB$^{Cy3B}$ experiment were divided into two disjoint subsets of RNAP binding events $\sigma^{70}$RNAP$^{647}$ that arrived at DNA 1) without GreB$^{Cy3B}$ and 2) as a $\sigma^{70}$RNAP$^{647}$•GreB$^{Cy3B}$ complex (**Figure 3—figure supplement 3**).

entire interval during which RNAP was bound to DNA. Taken together, these observations strongly suggest that $\sigma^{70}$RNAP molecules that are complexed with GreB cannot initiate transcription on *rrnB* P1, and again show that GreB blockage of the *rrnB* initiation pathway occurs at a step after formation of the initial closed complex intermediates.

## Discussion

Proteins that regulate transcription initiation can do so by affecting the initial binding of RNAP to the promoter or by modulating later step(s) such as promoter complex isomerization, initial RNA synthesis, or promoter escape (*Browning and Busby, 2016*). In this study, we show that GreB repression of initiation at a rRNA promoter falls into the latter category. We show by direct observation that GreB binds to free $\sigma^{70}$RNAP in solution and its presence on RNAP does not affect the formation of initial closed complexes, their stabilities, or their rates of dissociation (*Figure 5*). Instead, GreB acts at a later stage: closed complexes with GreB bound were never observed to progress to ECs, indicating that the secondary channel factor presence introduces a barrier to initiation at, for example, the closed-to-open complex isomerization stage or a subsequent step; this barrier is effectively insurmountable because it cannot be overcome on the time scale of *rrnB* P1 promoter complex lifetimes. Our data suggest that GreB and DksA inhibit rrnB by similar mechanisms, and the results are therefore compatible with proposals that DksA inhibits rRNA synthesis at least in part by affecting isomerization or open complex decay (*Paul et al., 2005*; *Rutherford et al., 2009*; *Rutherford et al., 2007*). The data are also consistent with a mechanism inspired by structural data (e.g., *Molodtsov et al., 2018*) in which the two SCFs may additionally inhibit initiation by obstructing catalytic function of the open complex active site.

Perhaps unexpectedly for a factor affecting only later stage(s) of initiation, we find that GreB does not function by binding to *rrnB* promoter complexes during those stages. In our experiments, GreB was almost never observed to bind to or depart from $\sigma^{70}$RNAP-promoter complexes. Instead, GreB arrived only at promoters that were not already bound by $\sigma^{70}$RNAP and arrived only in complex with $\sigma^{70}$RNAP. Similarly, it left the promoter only as a $\sigma^{70}$RNAP•GreB complex. However, rRNA promoter complexes are atypically short lived (*Ruff et al., 2015*). On more typical promoters, promoter complexes may be sufficiently long-lived to allow association of and dissociation of GreB. Dissociation of

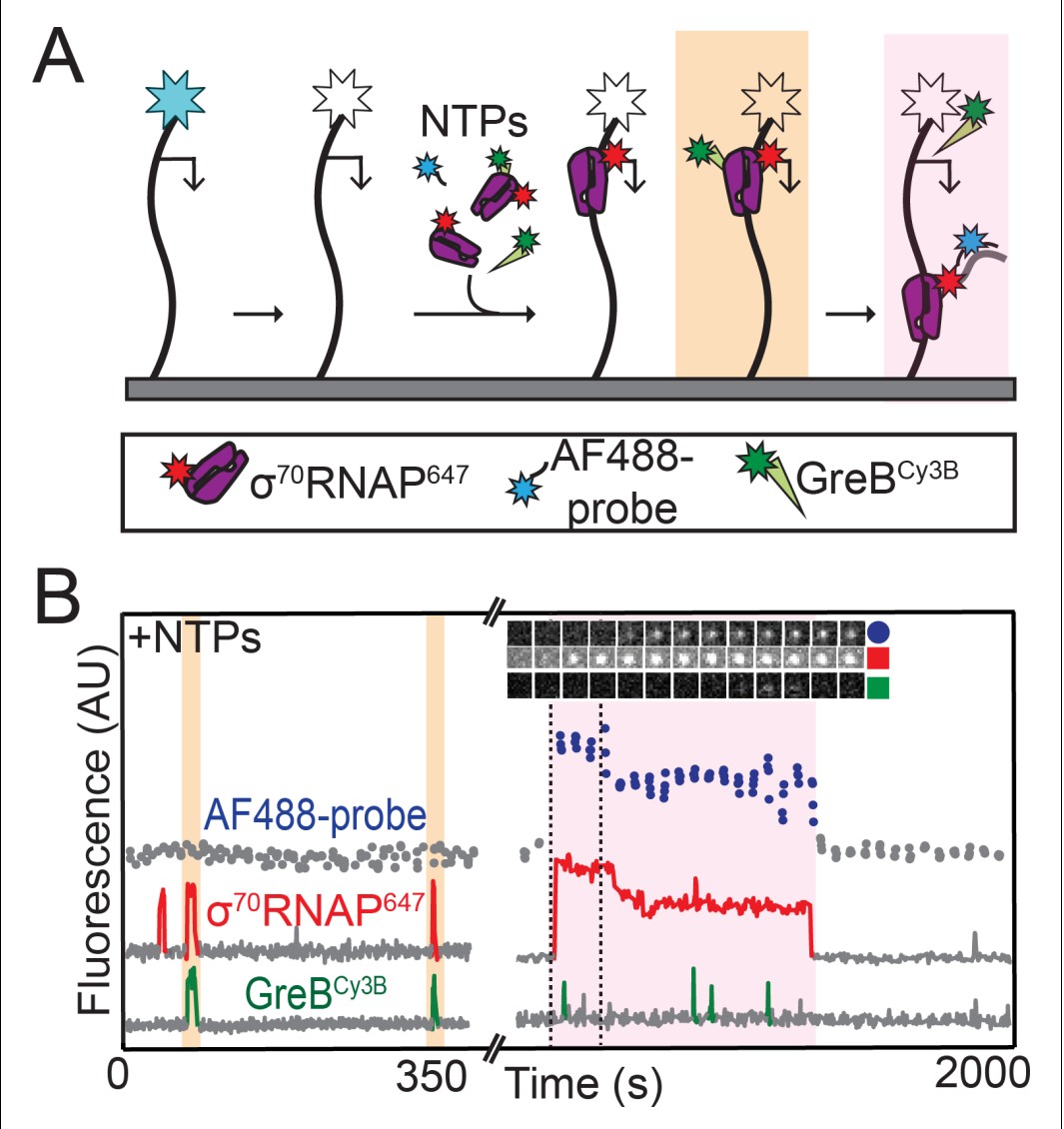

**Figure 4.** RNAP does not initiate transcription when GreB is bound. (**A**) Experiment design. Template and labeled proteins were the same as in *Figure 3A*. DNA was first imaged, then the AF488 dye (blue star) was bleached (white star). Then AF488 oligonucleotide probes and 1 mM NTPs were introduced as in *Figure 2*, along with 1 nM σ70RNAP647 and 10 nM GreBCy3B. (**B**) Two excerpts from the record of fluorescence intensities colocalized with one template DNA molecule. Two events in which σ70RNAP647 bound to DNA accompanied by GreBCy3B (orange) and one in which σ70RNAP647 arrived alone (purple) are highlighted. Only the last resulted in transcription initiation as judged by AF488-probe colocalization. Time series images (1.1 × 1.1 μm) from the time interval delimited by dashed lines confirm appearance of co-localized AF488-probe and σ70RNAP647.

The online version of this article includes the following figure supplement(s) for figure 4:

**Figure supplement 1.** Additional fluorescence intensity records demonstrating σ70RNAP647, GreBCy3B, and AF488-oligo probe co-localization at single DNA locations in the presence of NTPs, as in *Figure 4B*.

---

GreB during a long-lived promoter complex would prevent the suppression of initiation by the mechanism we observe with *rrnB* P1, providing a straightforward explanation for why GreB (and DksA) selectively inhibits initiation at promoters that have short-lived promoter complexes (*Paul et al., 2005*; *Rutherford et al., 2007*). On promoters with long-lived RNAP complexes, inhibition by this mechanism would be abolished or weakened, and stimulatory effects of Gre factors (e.g., preventing abortive initiation [*Chander et al., 2015*; *Duchi et al., 2016*; *Feng et al., 1994*; *Hsu et al., 1995*;

*Lerner et al., 2016*] or suppressing promoter-proximal pausing [*Deighan et al., 2011*; *Marr and Roberts, 2000*]) observed at some promoters could act to accelerate initiation.

The demonstration that GreB occupancy of holoenzyme is what dictates GreB inhibition of *rrnB* suggests that competition between SCFs for binding to σ[70]RNAP in solution (i.e., before it binds to promoter) may determine which SCFs are most important for rRNA transcription regulation under various growth conditions. Here we used three different methods to assess the affinity with which GreB binds to both core RNAP and σ[70]RNAP. First, we used FCS to measure the affinity of GreB for core and σ[70]RNAP (*Figure 1* and *Table 1*). We then confirmed this affinity measurement using a single-molecule assay in which we also measured the kinetic parameters of GreB binding to and dissociating from surface-tethered core and σ[70]RNAP (*Figure 1—figure supplement 2* and *Table 1*). Finally, we measured the fraction of σ[70]RNAP molecules that arrive at and depart from surface-tethered DNA containing the *rrnB* P1 promoter in complex with GreB (*Figure 3C*). All three approaches showed that that GreB binds σ[70]RNAP with a $K_D$ more than 10-fold tighter than its binding to ECs (*Table 1*) and much tighter than prior estimates based on non-equilibrium studies of its affinity for core RNAP or initial transcription complexes (*Furman et al., 2013*; *Lennon et al., 2012*; *Lennon et al., 2009*; *Loizos and Darst, 1999*; *Rutherford et al., 2007*). The tight binding affinity of GreB for holoenzyme is also consistent with the observed potent inhibition of rrnB P1 initiation kinetics by low concentrations (10 nM) of GreB. Since the binding of GreB to holoenzyme is much tighter than to ECs, holoenzyme molecules are likely able to at least partially compete with ECs for association with unbound GreB, even if the latter are present in significant excess.

Our observation that GreB binds to and releases from holoenzyme, not rRNA promoter complexes, provides strong support for the proposal (*Rutherford et al., 2007*) that the *relative* importance of GreB and DksA in regulation of initiation at *rrnB* P1 is simply determined by which one dominates occupation of the secondary channel in the cellular pool of σ[70]RNAP. Equilibrium occupancy is determined by both the binding affinities (measured here for GreB) and the ratios of unbound (not total) SCF concentrations, which are not known. The two proteins may have redundant functions in growth-rate dependent rRNA regulation, but the phenotypes of genetic deletion strains (*Rutherford et al., 2007*) are difficult to interpret unambiguously because of the other, non-overlapping functions of the two proteins in transcription elongation and other cellular processes (*Furman et al., 2013*; *Furman et al., 2012*; *Tehranchi et al., 2010*; *Zhang et al., 2014*). In contrast, the millimolar levels of ppGpp present during the stringent response may selectively promote DksA binding over that of GreB (*Molodtsov et al., 2018*; *Ross et al., 2016*). Thus, DksA may become the dominant regulator of the two during amino acid starvation. Further study will be required to

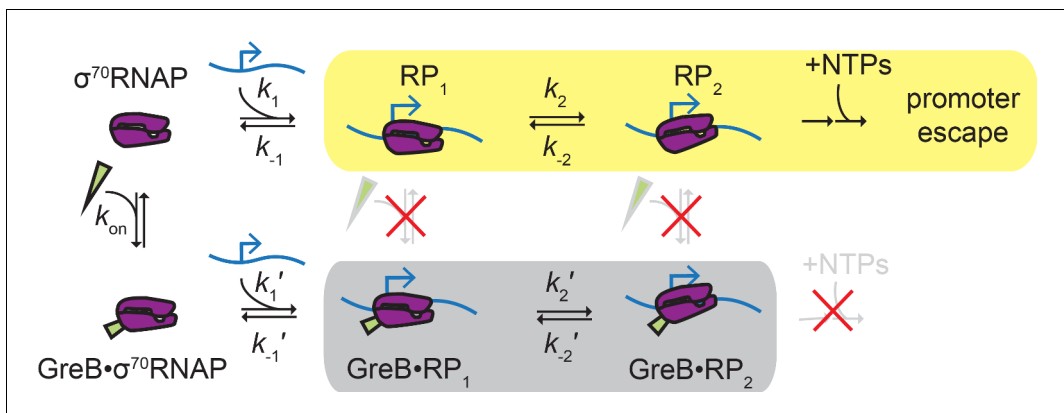

**Figure 5.** Proposed 'delayed inhibition' mechanism for growth-dependent regulation of rRNA initiation. σ[70]RNAP (purple) binds promoter DNA (blue) and forms closed complexes $RP_1$ and $RP_2$ before isomerization to open complex, initial transcript synthesis, and promoter escape in the productive initiation pathway (yellow). GreB (green) binds rapidly and reversibly to free σ[70]RNAP (*Table 1*), and GreB does not alter the kinetics of closed complex formation or decay (*Table 4*). However, GreB blocks a later step in initiation and GreB dissociation from RNAP cannot compete with RNAP dissociation from the promoter; thus, promoter binding by GreB•σ[70]RNAP is a non-productive pathway (gray) that does not initiate RNA synthesis. DksA likely regulates rRNA initiation by a similar mechanism.

**Table 4.** Rate constants (±SE*) for kinetic scheme depicted in **Figure 5**.

| | $k_1$ or $k_1'$ (M$^{-1}$s$^{-1}$)[†] | $k_{-1}$ or $k_{-1}'$ (s$^{-1}$)[‡] | $k_2$ or $k_2'$ (s$^{-1}$)[‡] | $k_{-2}$ or $k_{-2}'$ (s$^{-1}$)[‡] |
|---|---|---|---|---|
| σ$^{70}$RNAP | $2.2 \pm 0.6 \times 10^6$ | $0.48 \pm 0.12$ | $1.4 \pm 0.6 \times 10^{-2}$ | $2.4 \pm 0.8 \times 10^{-2}$ |
| σ$^{70}$RNAP·GreB | $2.4 \pm 0.7 \times 10^6$ | $0.30 \pm 0.16$ | $2.3 \pm 0.7 \times 10^{-2}$ | $1.7 \pm 0.4 \times 10^{-2}$ |

*Determined by bootstrapping (**Friedman and Gelles, 2012**).

[†]Identical to the corresponding $k_a$ values in **Table 3**.

[‡]Calculated as described (**Friedman and Gelles, 2012**) from the dwell time fit parameters reported in **Table 3**.

determine the extent that competition with other SCFs (e.g., GreA) affects the function of GreB and DksA in rRNA regulation.

Interestingly, the interaction of GreB with free RNAP may be more complex than a simple one-step binding mechanism. Single-molecule kinetic measurements show a biexponential dwell time distribution of GreB on surface-tethered core and σ$^{70}$RNAP (**Figure 1—figure supplement 2D,F**), indicating distinct short- and long-lived classes of binding events. Only a very small fraction of GreB interactions with holoenzyme (11%; calculated as (1 − $a$) from the value of $a$ in **Table 1**) achieve the long-lived state, and what causes the different behaviors of the two classes is obscure.

All stages of the transcription cycle are thought to be modulated by SCFs, and each SCF is presumed to compete for the same binding site on RNAP. Therefore, knowing the steps in the transcription mechanism at which SCFs load onto and dissociate from RNAP, as well as the dynamics of this interaction and the manner in which SCF presence affects the kinetic properties of RNAP while bound, can illuminate how these factors are able to exert specific regulatory effects at precise stages in the cycle. Using GreB as a model SCF, we propose based on our data that transcription factor binding to free σ$^{70}$RNAP, rather than to initiation complexes, is the critical factor that determines the relative importance of each SCF in inhibiting initiation from rRNA promoters. The data reveal that GreB functions by a 'delayed inhibition' mechanism, and imply that DksA functions by the same mechanism. Based on this mechanism, we propose that rRNA promoters are selectively inhibited by SCFs because their short-lived RNAP complexes do not allow sufficient time for SCFs to dissociate.

# Materials and methods

## Key resources table

| Reagent type (species) or resource | Designation | Source or reference | Identifiers | Additional information |
|---|---|---|---|---|
| Software | CoSMoS analysis software | https://github.com/gelles-brandeis/CoSMoS_Analysis | RRID:SCR_01689 | |

### Hybridization probe oligonucleotides

Transcription initiation was detected by hybridization of two fluorescently-labeled oligonucleotide probes to the nascent transcript. Probes were fluorescently labeled with either 5(6)-Carboxytetramethylrhodamine (TAMRA) (5′-/56-TAMN/gcgtgtttgccgttgttccgtgtcagt-3′ and 5′-/56-TAMN/tctcaggagaaccccgctgacccggcg-3′), or with AlexaFluor488 (AF488) (5′-/5Alex488N/gcgtgtttgcc gttgttccgtgtcagt-3′ and 5′-/5Alex488N/tctcaggagaaccccgctgacccggcg) (all from IDT DNA).

### rrnB P1 transcription template

The 314 bp AF488-labeled transcription template consisted of the *E. coli rrnB* gene from positions −65 to +249 (relative to the transcription start site defined as +1) and included the full *rrnB* P1 including the upstream promoter (UP) element. The template was synthesized by PCR using AF488- and biotin-labeled primers 5′-/AF488/GCGGTCAGAAAATTATTTTAAATTTCC-3′ and 5′-/5Biosg/CGTGTTCACTCTTGAGACTTGGTATTC-3′ (IDT DNA), *E. coli* genomic DNA from strain MG1655 (ATCC), and Herculase II Fusion DNA polymerase (Agilent) according to the manufacturer's protocol.

## Synthesis of BG-488-PEG-biotin

The bifunctional benzylguanine (BG) adduct BG-488-PEG-biotin (which includes both an ATTO 488 dye and a biotin moiety) was prepared by coupling of synthetic intermediate BG-Lys(NH$_2$)-PEG-biotin and commercially available ATTO 488 NHS ester (ATTO-Tec GmbH, Siegen, Germany). The synthetic intermediate BG-Lys(NH$_2$)-PEG-Biotin was obtained from successive couplings of commercially available α-N-Fmoc-ε-N-Dde-lysine (Merck KGaA, Darmstadt, Germany) with BG-NH$_2$ (New England Biolabs, Ipswich, MA, USA) and N-(+)-biotin-6-aminocaproic acid N-succinimidyl ester (Sigma-Aldrich, St. Louis, MO) by the synthetic route described previously (*Smith et al., 2013*). BG-488-PEG-biotin was synthesized as follows: BG-Lys(NH$_2$)-PEG-biotin (2.4 mg, 2.6 µmol) was dissolved in anhydrous DMF (1 mL). ATTO 488 NHS ester (2.8 mg, 2.9 µmol) and triethylamine (1.0 µL, 7.3 µmol) were added and the reaction mixture stirred overnight at room temperature. The solvent was removed under vacuum and the product purified by reversed-phase HPLC on a VYDAC 218TP series C18 column (22 × 250 mm, 10 µm particle size) at a flow rate of 20 mL/min using a 0.1 M triethyl ammonium bicarbonate/acetonitrile gradient. HPLC fractions containing the product were lyophilized to afford 1.13 µmol (43% yield) of BG-488-PEG-biotin. A high-resolution mass spectrum of BG-488-PEG-biotin was recorded by electrospray ionization (ESI) on an Agilent 6210 Time-of-Flight (TOF) mass spectrometer: ESI-TOFMS *m/z* 756.7879 [M + 2H]$^{2+}$ (calc. for C$_{69}$H$_{89}$N$_{15}$O$_{18}$S$_3$, *m/z* 756.7909).

**Scheme 1.** Structure of BG-488-PEG-biotin.

## Labeled *E. coli* RNAP preparations

*E. coli* core RNAP ($\alpha_2\beta\beta'\omega$) with a SNAP-tag on the C-terminus of β' (*Tetone et al., 2017*) was labeled with SNAP-Surface 647 (New England Biolabs) as described (*Mumm et al., 2020*) to produce RNAP$^{647}$. Labeling stoichiometry measured as in *Mumm et al. (2020)* was 54%. RNAP$^{488\text{-PEG-biotin}}$ was prepared in the same way, except that BG-488-PEG-biotin was used and unreacted dye was removed using a 30 kDa MWCO centrifugal filter (Microcon MRCF0R030; EMD Millipore); labeled protein was subjected to two cycles in which the protein was concentrated ~4 fold and then re-diluted to its original volume. To prepare $\sigma^{70}$RNAP$^{647}$ or $\sigma^{70}$RNAP$^{488\text{-PEG-biotin}}$, labeled core RNAP was incubated with $\sigma^{70}$ as described (*Harden et al., 2016*).

## GreB preparation and fluorescent labeling

GreB proteins used in this study were either wild type ('GreB') or contained a relocated cysteine ('rcGreB'; E82C/C68S mutant) and were purified as previously described (*Tetone et al., 2017*). rcGreB was labeled with either Cy3- or Cy3B-maleimide monoester (GE Healthcare, Piscataway, NJ) as previously described (*Tetone et al., 2017*). GreB$^{Cy3B}$ was determined to be 93% labeled as described previously (*Tetone et al., 2017*) and GreB$^{Cy3}$ preparations were determined to be 80–90% labeled as described (*Tetone et al., 2017*) with the exception that $\varepsilon_{550,Cy3}$ = 150,000 M$^{-1}$ cm$^{-1}$. GreB$^{Cy3}$ is fully active in transcript cleavage assays (*Tetone et al., 2017*) and its binding to

$\sigma^{70}$RNAP was quantitatively indistinguishable from that of unlabeled GreB in competition experiments (*Figure 1—figure supplement 2C*).

## Fluorescence Correlation Spectroscopy and autocorrelation analysis

All FCS data was collected in a custom one-photon confocal instrument (*Liu et al., 2006*; *Pu et al., 2009*). In brief, the 514 nm line was selected from an air-cooled multiline argon-krypton ion laser (Melles-Griot) using a prism and used to excite the sample. A 535drlp dichroic mirror (Chroma Technology) reflected the laser light into a water immersion objective (Olympus, NA = 1.2) mounted on an inverted microscope (IX-70, Olympus); the objective also collected the emitted fluorescence. The same dichroic passed the fluorescence emission, and any remaining scattered laser light was blocked by an HQ540lp filter (Chroma Technology). A 30 μm confocal pinhole (Thorlabs) in the conjugate image plane was used to define the observation volume and to block out-of-focus fluorescence. The fluorescence was collimated, split by a 635dcxr dichroic mirror (Chroma Technology) and the reflected fluorescence passed through an HQ580/30 bandpass filter (Chroma Technology) before being focused unto an avalanche photodiode (APD, SPCM-AQR-14, Perkin-Elmer). The transmitted fluorescence passed through a D675/50 bandpass filter (Chroma Technology) to block Raman scattering before being focused on the second APD. The photon counts from the APDs were collected by a 2-channel data acquisition card and the auto- and cross-correlation curves were calculated by associated software (ISS). The ISS software was also used to analyze the FCS data. Excitation power was 30–40 μW. Data acquisition was at $24 \times 10^{6}$ Hz and $1–2 \times 10^{6}$ photons were collected per run.

The beam width ($2\omega_o$) and height ($2z_o$) were determined by acquiring data with a standard dye, rhodamine-6G (Rh6G) using a diffusion coefficient, $D_{Rh6G}$, of 280 μm$^2$/s (*Magde et al., 1974*). Pulsed field gradient NMR experiments suggest that $D_{Rh6G}$ may be closer to 400 μm$^2$/s (*Gendron et al., 2008*). An increased value of $D_{Rh6G}$ has no effect on our determinations of the fraction of bound GreB ($f_{bound}$) or the associated dissociation constant ($K_D$), but could result in underestimation of the diffusion coefficients of free and bound GreB. For all FCS acquisitions with GreB$^{Cy3}$, $\omega_o$ and $S = z_o/\omega_o$ were held constant when fitting the autocorrelation traces.

For equilibrium binding measurements, fluorescence fluctuation data was first collected for GreB$^{Cy3}$ in assay buffer [40 mM Tris-Cl, pH 8.0, 130 mM KCl, 4 mM MgCl$_2$, 0.1 mM EDTA, 0.1 mM DTT, 10 mg/mL BSA] supplemented with oxygen scavenging reagents (*Friedman et al., 2006*). Autocorrelation of the fluorescence signal was calculated using ISS-Vista software for FCS analysis. Assuming that GreB$^{Cy3}$:RNAP and GreB$^{Cy3}$:$\sigma^{70}$RNAP binding is 1:1 and that the brightness (quantum yield) of Cy3 does not change upon binding, the correlation curves, $G(\tau)$ may be fit to

$$G(\tau) = \frac{1}{\left(\pi\sqrt{\pi}\omega_o^3 S\langle C\rangle\right)} \sum_i f_i \left(\left(1 + \frac{4D_i\tau}{\omega_o^2}\right)\sqrt{1 + \frac{4D_i\tau}{S^2\omega_o^2}}\right)^{-1} \tag{1}$$

where $<C>$ is the time averaged concentration in the observation volume, $f_i$ is fraction of species $i$ and $D_i$ is the diffusion coefficient of species $i$. The diffusion coefficient of GreB$^{Cy3}$ ($D_{GreB}$) was obtained by fixing $f = 1$.

Core RNAP or $\sigma^{70}$RNAP was then added at a concentration ($R_T$) ranging from 1 to 100 nM to a fixed $C = 10$ nM concentration of GreB$^{Cy3}$. Fluorescence records were collected at each RNAP concentration, and the autocorrelation was obtained as described above. Curves were then fit to *Equation 1* for two species with fixed $D_{GreB}$ to obtain $N$, $f_{bound}$, and the diffusion constant of the GreB·RNAP complex $D_{GreB·RNAP}$. Standard errors of the fit parameters were obtained by bootstrapping.

The concentration of RNAP bound, $R_{bound}$, is

$$R_{bound} = C \times f_{bound}. \tag{2}$$

Therefore, concentration of free RNAP, $R_{free}$, was calculated as

$$R_{free} = R_T - R_{bound}. \tag{3}$$

$K_D$ was measured by performing a standard error weighted fit of the pairs of $f_{bound}$ and $R_{free}$ values from measurements at different $R_T$ to the Langmuir equation

$$f_{\text{bound}} = [\text{R}_{\text{free}}]/([\text{R}_{\text{free}}] + \text{K}_{\text{D}}). \tag{4}$$

## CoSMoS

Single-molecule fluorescence microscopy experiments were conducted using a multi-wavelength micromirror total internal reflection fluorescence microscope (*Friedman et al., 2006*) equipped with lasers at wavelengths 488, 532, and 633 nm for fluorescence excitation and at 785 nm for autofocus (*Friedman and Gelles, 2015*). Experiments were conducted in assay buffer supplemented with an oxygen scavenging system (*Friedman et al., 2006*). All experiments were conducted at 20–23° C. Data were acquired at 0.5 s per frame except as otherwise noted.

CoSMoS experiments were conducted in flow chambers (*Friedman et al., 2006*) using glass slides passivated with a mPEG-SG2000:biotin-PEG-SVA5000 (Laysan Bio) 200:1 w/w mixture (*Friedman et al., 2013*). Streptavidin-coated fluorescent beads (T-10711, Molecular Probes) were added to the chamber at a dilution of ~1:400,000 to serve as fiducial markers for drift correction (*Friedman and Gelles, 2015*). The chamber was then incubated with 0.013 mg/mL NeutrAvidin (31000, Thermo Fisher) in assay buffer for 45 s and then was flushed with three to five chamber volumes (20 µl) of assay buffer.

Initiation experiments were performed essentially as described previously (*Mumm et al., 2020*) except that data were acquired at 1 s per frame.

Equilibrium binding of GreB[Cy3B] to surface-tethered RNAP was measured by incubating the chamber with ~20 pM RNAP[488-PEG-biotin] until a sufficient surface density of fluorescent spots (*Friedman and Gelles, 2015*) was obtained. The chamber was then flushed with buffer, and images of a new field of view were recorded to establish locations of the RNAP[488-PEG-biotin] molecules. GreB[Cy3B] was then introduced and the association to and dissociation from surface-tethered RNAP[488-PEG-biotin] molecules was monitored.

## CoSMoS data analysis

Image and kinetic analysis was performed using custom software and algorithms for automatic spot detection, spatial drift correction, colocalization, plotting and analyzing kinetic data, and determining association and dissociation rates, active fractions, and amplitudes (https://github.com/gelles-brandeis/CoSMoS_Analysis) (*Friedman and Gelles, 2015*). The accompanying data archive (doi: 10.5281/zenodo.2530159) provides single-molecule source data for each figure and figure supplement; the files are 'intervals' files readable by the imscroll program in the Github repository cited above. The rates of association and dissociation of GreB[Cy3B] to surface-tethered σ70RNAP[488-PEG-biotin] were measured as described previously (*Tetone et al., 2017*), except that the selected locations on the slide surface were tethered RNAP[488-PEG-biotin] molecules instead of DNA molecules.

The rate of transcription initiation on *rrnB* P1 templates tethered to the chamber surface was measured as described previously (*Friedman and Gelles, 2015*; *Mumm et al., 2020*). Initiation rates ($k_{\text{init}}$) were corrected for the rate of nonspecific TAMRA-probe or AF488-probe binding ($k_{\text{bkgnd}}$) determined by analyzing randomly selected surface locations without AF488-labeled DNA molecules from the same recording. In *Figure 2C–F*, experiments without SCF (purple curves) were used to obtain the active fraction as described (*Friedman and Gelles, 2015*; *Friedman and Gelles, 2012*). The corresponding experiments with SCF and SCF+ppGpp were performed on the same day and the active fraction $A_{\text{f}}$ of these curves was fixed to the value obtained in the experiment without SCF.

The association rate of σ70RNAP[647] to *rrnB* P1 DNA locations in the absence and presence of SCFs was determined in the absence of NTPs essentially as described previously (*Mumm et al., 2020*), with the exception that the rate of background binding was determined by measuring the rate of non-specific binding to the chamber surface rather than to DNA templates lacking the *rrnB* P1 promoter. Fitting of the time to first binding distributions yielded values (*Table 3*) for the second-order association rate constant for non-specific surface binding ($k_{\text{bkgnd}}$), the second-order association rate constant for promoter DNA binding ($k_{\text{a}}$), and the fraction of DNA molecules that were active in binding ($A_{\text{f}}$) (*Friedman and Gelles, 2015*). The dissociation of σ70RNAP[647] from *rrnB* P1 DNA was measured essentially as previously described (*Mumm et al., 2020*). Photobleaching contributions for σ70RNAP[647] dwell times on *rrnB* P1 DNA and GreB[Cy3B] on surface-tethered σ70RNAP or RNAP core determined as described previously (*Hoskins et al., 2016*) and were negligible.

## Acknowledgements

We thank Larry Tetone and David Harbage for help with protein purification and labeling and thoughtful discussion. This work was supported by NIH grants R01GM081648 (JG) and R01GM38660 (RL) and NSF grant MCB-0446220 (AG).

## Additional information

### Competing interests

Ivan R Corrêa Jnr: Employed by New England Biolabs. The author has no other competing interests to declare. The other authors declare that no competing interests exist.

### Funding

| Funder | Grant reference number | Author |
| --- | --- | --- |
| National Institute of General Medical Sciences | R01GM081648 | Jeff Gelles |
| National Science Foundation | MCB-0446220 | Anne Gershenson |
| National Institute of General Medical Sciences | R01GM38660 | Robert Landick |

The funders had no role in study design, data collection and interpretation, or the decision to submit the work for publication.

### Author contributions

Sarah K Stumper, Conceptualization, Resources, Data curation, Software, Formal analysis, Investigation, Methodology, Writing—original draft, Project administration, Writing—review and editing; Harini Ravi, Conceptualization, Formal analysis, Investigation, Methodology, Writing—original draft, Writing—review and editing; Larry J Friedman, Software, Formal analysis, Methodology, Writing—review and editing; Rachel Anne Mooney, Resources, Writing—review and editing; Ivan R Corrêa Jnr, Resources, Writing—original draft, Writing—review and editing; Anne Gershenson, Conceptualization, Formal analysis, Supervision, Methodology, Writing—review and editing; Robert Landick, Conceptualization, Resources, Writing—review and editing; Jeff Gelles, Conceptualization, Formal analysis, Supervision, Funding acquisition, Writing—original draft, Writing—review and editing

### Author ORCIDs

Larry J Friedman http://orcid.org/0000-0003-4946-8731
Anne Gershenson http://orcid.org/0000-0003-3124-9610
Robert Landick https://orcid.org/0000-0002-5042-0383
Jeff Gelles http://orcid.org/0000-0001-7910-3421

### Decision letter and Author response

Decision letter https://doi.org/10.7554/eLife.40576.sa1
Author response https://doi.org/10.7554/eLife.40576.sa2

## Additional files

### Supplementary files

• Transparent reporting form

### Data availability

All data analyzed for this study are included in the manuscript or in the source data files (https://doi.org/10.5281/zenodo.2530159).

The following dataset was generated:

| Author(s) | Year | Dataset title | Dataset URL | Database and Identifier |
|---|---|---|---|---|
| Gelles J | 2019 | Single-molecule source data files from Delayed inhibition mechanism for secondary channel factor regulation of ribosomal RNA transcription | https://doi.org/10.5281/zenodo.2530159 | Zenodo, 10.5281/zenodo.2530159 |

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
