## [Decision Letter]

Thank you for submitting your article "Delayed inhibition mechanism for secondary channel factor regulation of ribosomal RNA transcription" for consideration by *eLife*. Your article has been reviewed by Kevin Struhl as the Senior Editor, a Reviewing Editor, and two reviewers. The reviewers have opted to remain anonymous.

The reviewers have discussed the reviews with one another and the Reviewing Editor has drafted this decision to help you prepare a revised submission. As you can see from their comments, both reviewers are enthusiastic about the work. None of the issues they raise were seen as critical to the study, but would strengthen the manuscript if addressed.

Summary:

Stumper et al. use single-molecule fluorescence experiments to study RNAP control through the secondary channel, providing a detailed view of the regulatory mechanism that could not be obtained from bulk experiments. The work is an elegant demonstration of the inhibitory role of GreB in transcription initiation from the classical *rrnB* P1 promoter. Using FCS and single-molecule surface assays the authors show that GreB binding to core or holoenzyme RNAP is tighter than previously thought (dissociation constant 10 nM). Using multiwavelength single-molecule assays the authors then demonstrate identical binding of both RNAP holoenzyme and RNAP holoenzyme complexed with GreB to the *rrnB* P1 promoter in the absence of nucleotides, and complete inhibition of escape from the *rrnB* P1 promoter of GreB-bound RNAP holoenzyme. The authors also examine the similar role of DksA and the synergistic, inhibitory role of ppGpp.

Essential revisions:

None

Minor points:

1) The authors extend their conclusions to other SC factors, arguing that they bind to the enzyme similarly and all effects will be controlled by occupancy. It could be more interesting to point out differences and it is possible that some SC factors are not exchanging that rapidly. In their recent study (Tetone et al., 2017), the authors found that a cleavage-deficient D41N GreB remained bound to the elongation complex much longer. A "native" Gre-like Gfh, which inhibits catalysis, could have similar properties.

2) The authors stress the point that GreB arrives and departs with the holoenzyme, instead of binding to the initiation complexes. This could in part be explained by the low stability of these complexes, and maybe this point could be made clearer. Alternatively, this result may suggest that GreB affinity for the DNA-bound RNAP is lower. Experimentally, this could be tested by including ATP which is expected to stabilize an otherwise absent open complex. This may also allow distinction between GreB effects on open complex vs nucleotide addition.

3) While GreB is an excellent model to study control through the SC, the manuscript may convey a notion that GreB makes an important contribution to regulation of rRNA and perhaps other promoters that form very short-lived complexes. This scenario appears to be unlikely in the cell: unless the authors could cite evidence to the contrary, *E. coli* GreB is present in very low numbers (near K_D_ determined here) and would not be able to shift the balance, particularly in the case of rRNA synthesis carried out by numerous RNAPs. Ten RNAP:GreB nonproductive complexes would not shift the balance – unless GreB is overexpressed under some yet TBD conditions. Even a much more abundant GreA affects initiation from just a handful of promoters. The authors should explicitly discuss this issue and cite known in vivo GreB concentrations.

4) It would be helpful for the authors to comment on the results they presented in Figure 1D of Tetone et al. (2016), in which the PR' promoter was used and RNAP was in the presence of 1 μM of GreB and relate them to this work. In Tetone et al. the authors were able to observe RNA polymerases transcribing, yet according to the arguments here the polymerase is likely saturated with GreB and arrives at PR' promoter with GreB. Should readers expect that GreB dissociates from polymerase bound to the PR' promoter and that this allows transcription to take place, or that polymerase can escape from PR' with GreB bound?

5) In Figure 2G, I was expecting a 50% inhibition of transcription initiation given that GreB was at 10 nM (where polymerase is 50% occupied) but it looks to be closer to 75% inhibition. Is this interpretation correct?

6) It was somewhat unclear on what basis the authors assert which states are accessible to the polymerase bound to *rrnB* P1 in the absence of nucleotides.

7) There are only a couple of places that should be in italics = *E. coli* (Subsection “Fluorescence Correlation Spectroscopy and Autocorrelation Analysis”) and rrnB (subsection “GreB arrives at and departs from an rRNA promoter in complex with σ70RNAP”) are not in italics and a space is missing between # and nM (Figure 3 legend).

---

## [Author Response]

Essential revisions:NoneMinor points:1) The authors extend their conclusions to other SC factors, arguing that they bind to the enzyme similarly and all effects will be controlled by occupancy. It could be more interesting to point out differences and it is possible that some SC factors are not exchanging that rapidly. In their recent study (Tetone et al., 2017), the authors found that a cleavage-deficient D41N GreB remained bound to the elongation complex much longer. A "native" Gre-like Gfh, which inhibits catalysis, could have similar properties.

We intended our conclusions to apply only to GreB and DksA. We agree that other SC factors may function differently (including possibly having little or no effect on rRNA initiation). We have revised the text of the Discussion section to clarify these points and to explicitly say that understanding the role of other SC factors will require further work. We also changed one sentence in the abstract to avoid the unintended implication that the results apply generally to all SC factors.

2) The authors stress the point that GreB arrives and departs with the holoenzyme, instead of binding to the initiation complexes. This could in part be explained by the low stability of these complexes, and maybe this point could be made clearer. Alternatively, this result may suggest that GreB affinity for the DNA-bound RNAP is lower. Experimentally, this could be tested by including ATP which is expected to stabilize an otherwise absent open complex. This may also allow distinction between GreB effects on open complex vs nucleotide addition.

We agree with the first possible explanation suggested by the reviewers, and we have edited the Discussion section to make this clearer; the relevant part now reads:

“In our experiments, GreB was almost never observed to bind to or depart from σ^70^RNAP-promoter complexes. Instead, GreB arrived only at promoters that were not already bound by σ^70^RNAP and arrived only in complex with σ^70^RNAP. Similarly, it left the promoter only as a σ^70^RNAP•GreB complex. However, rRNA promoter complexes are atypically short lived (Ruff et al., 2015). On more typical promoters, promoter complexes may be sufficiently long-lived to allow association of and dissociation of GreB. Dissociation of GreB during a long-lived promoter complex would prevent the suppression of initiation by the mechanism we observe with *rrnB* P1, providing a straightforward explanation for why GreB (and DksA) selectively inhibits initiation at promoters that have short-lived promoter complexes (Paul et al., 2005; Rutherford et al., 2007).”

We also agree with the second point (which is not mutually exclusive with the first), at least to the extent that GreB associating more slowly with promoter complexes than with σ^70^RNAP holoenzyme (and therefore having a lower affinity for the latter) may contribute to the failure of GreB to bind those complexes during initiation. If that were the case, it would further support our conclusion that GreB loads onto holoenzyme prior to initial binding of RNAP to the promoter.

We respectfully disagree with the suggestion that the ATP-only experiment would be informative. We are skeptical that an ATP-stabilized state is a good mimic of the kinetic properties of the open complex intermediate in initiation. To the extent that ATP-only conditions prolong the lifetime of an open complex-like structure, these conditions might artifactually make it appear that GreB can associate with the open complex intermediate whereas under transcription conditions (four NTPs) the lifetime of this intermediate is likely too short for appreciable GreB association, even if the association rate constant were as high as for GreB binding to holoenzyme.

3) While GreB is an excellent model to study control through the SC, the manuscript may convey a notion that GreB makes an important contribution to regulation of rRNA and perhaps other promoters that form very short-lived complexes. This scenario appears to be unlikely in the cell: unless the authors could cite evidence to the contrary, *E. coli* GreB is present in very low numbers (near K_D_ determined here) and would not be able to shift the balance, particularly in the case of rRNA synthesis carried out by numerous RNAPs. Ten RNAP:GreB nonproductive complexes would not shift the balance – unless GreB is overexpressed under some yet TBD conditions. Even a much more abundant GreA affects initiation from just a handful of promoters. The authors should explicitly discuss this issue and cite known in vivo GreB concentrations.

We agree that this is an interesting question; the problem is that there are widely differing literature reports of the cellular abundances of DksA and GreB. For example, a modern Super-SILAC mass spectrometry study (Soufi et al., 2015) reports both proteins sub-stoichiometric with RNAP, with DksA and GreB respectively at ~5% and ~2.5% mole ratios to RNAP subunit rpoC, with relatively small (~2-fold) changes during different growth phases. In contrast, an older analysis of these specific proteins by quantitative western blots (Rutherford et al., 2006) gave rather different results: ~1,140% and ~ 108% respectively, again reportedly largely independent of growth phase. Another recent mass spectrometry study (Schmidt et al., 2016) yielded a still different picture (for growth in glucose media) with DksA and GreB respectively ~210% and ~0.03% of rpoC. (To be fair, the second number was flagged as being comparatively less reliable due to the limited number of peptides scored as being unique to GreB.) Clearly, there are wide discrepancies between studies even when one is comparing concentration ratios of the same three proteins.

Nevertheless, even if DksA and GreB are in fact present at <10% mole ratios relative to RNAP, it is still possible, at least in principle, that these SC factors could efficiently regulate rRNA transcription under rapid growth conditions, in which more than half of the transcriptional output is rRNAs and tRNAs. This is because holoenzyme free in solution (as opposed to promoter complexes, elongation complexes, or RNAP non-specifically bound to DNA) is <~10% of total RNAP under these conditions.* Free holoenzyme is the only one of these species demonstrated to bind DksA and GreB with high affinity, and in our model (Figure 5) free holoenzyme is the only subpopulation of RNAP that GreB need bind to regulate *rrnB* P1 initiation. (*According to most studies; see for example Bakshi et al., 2013; Klumpp and Hwa, 2008 and Patrick et al., 2015 but in contrast see Stracy et al., 2015).

Given the uncertainties about the cellular concentrations of GreB and DksA (to say nothing of the additional uncertainties about how much of each protein is free in solution, rather than bound in complex with RNAP), we feel that the manuscript’s discussion of this issue is appropriately circumspect. We merely cite the equilibrium competition model proposed by Rutherford et al. and point out that in the framework of this model both binding affinities and free concentrations influence the relative importance of GreB and DksA in regulation. This model is applicable even if the regulators are substoichiometric to RNAP.

4) It would be helpful for the authors to comment on the results they presented in Figure 1D of Tetone et al., (2016), in which the PR' promoter was used and RNAP was in the presence of 1 μM of GreB and relate them to this work. In Tetone et al. the authors were able to observe RNA polymerases transcribing, yet according to the arguments here the polymerase is likely saturated with GreB and arrives at PR' promoter with GreB. Should readers expect that GreB dissociates from polymerase bound to the PR' promoter and that this allows transcription to take place, or that polymerase can escape from PR' with GreB bound?

In Figure 1 of Tetone et al., the authors allowed open complexes to form on PR' without nucleotides and in the absence of GreB. Transcription was then initiated by washing out the free holoenzyme and simultaneously introducing GreB, NTPs, and hybridization probe. In this design, promoter escape is largely complete before any of the complexes have had a chance to bind GreB.

5) In Figure 2G, I was expecting a 50% inhibition of transcription initiation given that GreB was at 10 nM (where polymerase is 50% occupied) but it looks to be closer to 75% inhibition. Is this interpretation correct?

Yes, that is the logical expectation. There are a number of possible explanations for this difference. For example, there might be a minority subpopulation of polymerase molecules that are inactive (i.e., incapable of binding GreB/DksA and incapable of initiating transcription); this would lead to somewhat greater than half inhibition under conditions when half of total polymerase molecules were occupied by DksA or GreB. This discrepancy does not alter the validity of the conclusion of this section of the manuscript, which is that “GreB and DksA have identical effects on the rate of the full initiation process when the effects are measured at the same fractional occupancy of σ70RNAP”.

6) It was somewhat unclear on what basis the authors assert which states are accessible to the polymerase bound to rrnB P1 in the absence of nucleotides.

To the presentation of Figure 3 we added citations of key studies on this point (Barker et al., 2001 and Rutherford et al., 2009) and also corrected an incorrect citation to the relevant review article (Haugen et al., 2008 vs. Haugen et al., 2008). We also made minor edits to the Discussion section to clarify this issue.

7) There are only a couple of places that should be in italics = *E. coli* (Subsection “Fluorescence Correlation Spectroscopy and Autocorrelation Analysis”) and rrnB (subsection “GreB arrives at and departs from an rRNA promoter in complex with σ70RNAP”) are not in italics and a space is missing between # and nM (Figure 3 legend).

Thanks; we have corrected those.